# Research on Kinematics Analysis and Trajectory Planning of Novel EOD Manipulator

Jianwei Zhao [1],[†] , Tao Han [1],[*],[†] , Xiaofei Ma [2],[†] , Wen Ma [3] , Chengxiang Liu [1] , Jinyu Li [1] and Yushuo Liu [1]

1  School of Mechanical Electronic & Information Engineering, China University of Mining and Technology-Beijing, Beijing 100089, China; zhaojianwei@cumtb.edu.cn (J.Z.); ZQT1900401022G@student.cumtb.edu.cn (C.L.); SQT2100402034@student.cumtb.edu.cn (J.L.); SQT2100402037@student.cumtb.edu.cn (Y.L.)
2  College of Mechanical & Electrical Engineering, Nanjing University of Aeronautics and Astronautics, Nanjing 210016, China; ma960621@foxmail.com
3  Beijing Special Engineering and Design Institute, Beijing 100028, China; mw926@sina.com
*  Correspondence: ZQT1900401010G@student.cumtb.edu.cn
†  These authors contributed equally as co-first authors.

**Abstract:** To address the problems of mismatch, poor flexibility and low accuracy of ordinary manipulators in the complex special deflagration work process, this paper proposes a new five-degree-of-freedom (5-DOF) folding deflagration manipulator. Firstly, the overall structure of the explosion-expulsion manipulator is introduced. The redundant degrees of freedom are formed by the parallel joint axes of the shoulder joint, elbow joint and wrist pitching joint, which increase the flexibility of the mechanism. Aiming at a complex system with multiple degrees of freedom and strong coupling of the manipulator, the virtual joint is introduced, the corresponding forward kinematics model is established by D–H method, and the inverse kinematics solution of the manipulator is derived by analytical method. In the MATLAB platform, the workspace of the manipulator is analyzed by Monte Carlo pseudo-random number method. The quintic polynomial interpolation method is used to simulate the deflagration task in joint space. Finally, the actual prototype experiment is carried out using the data obtained by simulation. The trajectory planning using the quintic polynomial interpolation method can ensure the smooth movement of the manipulator and high accuracy of operation. Furthermore, the trajectory is basically consistent with the simulation trajectory, which can realize the work requirements of putting the object into the explosion-proof tank. The new 5-DOF folding deflagration manipulator designed in this paper has stable motion and strong robustness, which can be used for deflagration during the COVID-19 epidemic.

**Keywords:** COVID-19; 5-DOF manipulator; forward and inverse kinematics analysis; trajectory planning; MATLAB simulation

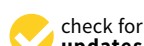



## 1. Introduction

In 2020, a sudden outbreak of the COVID-19 epidemic brought great changes to our lives. Presently, society is in the new normal of the epidemic. "Before the outbreak of the epidemic, terrorist attacks by extremist organizations were at most once a week, and now increased by at least five times, averaging more than 20 times a month," a senior Iraqi intelligence official [1] has revealed. Most of the terrorist attacks are explosion attacks, and the traditional manual explosive disposal methods cannot meet the needs of anti-terrorist war. Automatic explosive ordnance disposal (EOD) robots can replace humans to work in an unknown environment, including reconnaissance, detection, explosive disposal, transportation of explosives, and so on.

Now, some scientists have developed robots for dismantling explosive and burning devices. The British "trolley" [2,3] series, the German "TEODOR" [4] series and the American "Andros" [5,6] series have been developed and applied in battlefields or terrorist

activities. In China, there are also the "Lizards" [7,8] series of EOD robots, designed by the Institute of Automation of the Chinese Academy of Sciences, and the "Super-D" [9,10] series of EOD robots developed by Shanghai Jiao Tong University, equipped in various subway stations, parks and other densely populated areas.

Presently, the manipulators the EOD robots on the market are equipped with, such as ABB, YASKAWA and KUKA manipulators, are self-designed according to the robots' chassis and do not have universal application. Due to the design requirements of the project, for example, the weight of the mechanical arm is less than 50 kg, the arm span is required to be greater than or equal to 1000 mm and the maximum grasping capacity is greater than or equal to 10 kg. At present, most of the manipulators in the market do not match the mobile platform used in this paper, and their degrees of freedom and accuracy cannot meet the requirements of explosive removal. Aiming at the problems of mismatching poor flexibility and low precision of the ordinary manipulator in complex special explosive ordnance disposal processes, a new type of 5-DOF manipulator is proposed in this paper. The D–H model of the manipulator is established, and the forward and inverse kinematics are analyzed. The workspace of the manipulator is analyzed by Monte Carlo pseudo-random number method. MATLAB is used to simulate the trajectory planning, and the simulation data are used to verify the effectiveness of the design and analysis method.

The rest of this paper is arranged as follows: Section 2 briefly introduces the composition of the EOD robot and the design and analysis of the manipulator mechanism. Section 3 carries out the kinematics analysis of the manipulator. Section 4 simulates its workspace. Section 5 introduces the trajectory planning, simulation and experimental verification of the manipulator.

## 2. Mechanism Design

The main purpose of the robotic arm is to complete the two actions of grasping and placing explosives to realize the demolition of explosives [11]. The explosive disposal robot designed in this paper carries an explosion-proof device that can resist 400 g TNT. It needs to be equipped with a mechanical arm to place the explosives directly. In view of the mismatch between the ordinary manipulator and the mobile chassis and the problems of poor flexibility and low precision in the complex special explosive removal process, a new 5-DOF manipulator is proposed in this paper. To ensure the stability, reliability and flexibility of the manipulator, the choice of the number of degrees of freedom is particularly important when designing the mechanical structure. Because the manipulator is in-stalled on a mobile platform, the mobile platform can flexibly change the working position of the robot relative to the dangerous object, which is equivalent to a degree of freedom of rotation of the operating arm. Common industrial manipulators generally have 4–6 DOF. The higher the degree of freedom, the more flexible the manipulator is, but the greater the number of degrees of freedom, the more difficult it is to control the manipulator. In order to reduce the difficulty of control and complete the task, we chose 5 DOF. Therefore, the manipulator adopts the configuration of 5 + 1 DOF (5 DOF of the manipulator motion, 1 DOF of the end gripper opening and closing). The joints corresponding to the 5 DOF are the waist joint, shoulder joint, elbow joint, wrist pitch joint and wrist rotation joint. The joint axis of the shoulder joint, elbow joint and wrist pitch joint are parallel, thus forming a redundant degree of freedom, which can effectively increase the flexibility of the mechanism and improve the obstacle avoidance ability of the manipulator [12].

The configuration diagram of the manipulator is shown in Figure 1. Because the manipulator arm has 5 DOF and the waist joint is installed on the mobile chassis, the remaining 4 DOF are equivalent to three arms. Before the design, we also referred to the design parameters of ABB, YASKAWA and KUKA manipulators. Suppose the distance from the shoulder joint to the elbow joint is $L_1$, the distance from the elbow joint to the wrist pitch joint is $L_2$ and the distance from the wrist pitch joint to the end-effector is $L_3$. It is preset that the dead weight of the mechanical arm is 50 kg, the arm extension is 1.1m and the clamping weight is 5–10 kg. In this case, $L_1 + L_2 + L_3 \approx 1.1$ m. The numerical values of

$L_1$, $L_2$ and $L_3$ determine the working space of manipulator. In order to expand the working space of the manipulator as far as possible and reduce the working blind area, $L_3$ should be as small a value as possible, and the difference between $L_1$ and the $L_2$ should be as small as possible. At the same time, considering the dimensions after folding, the length of each part is set as follows: $L_1 = 0.5$ m, $L_2 = 0.37$ m, $L_3 = 0.30$ m.

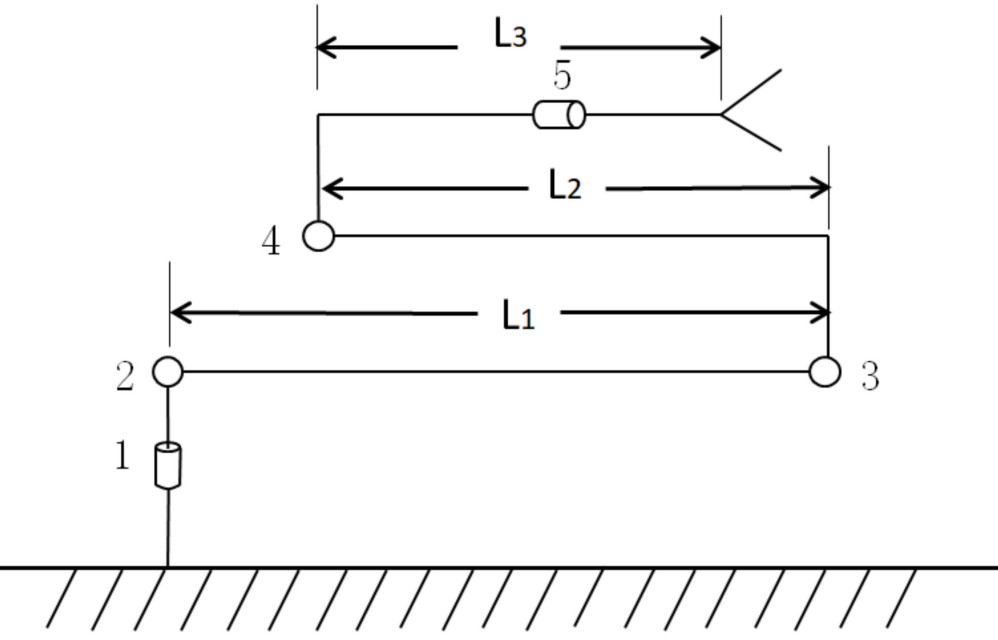

1-Waist joint  2-Shoulder joint  3-Elbow joint
4-Wrist pitch joint  5-Wrist rotation joint

**Figure 1.** Manipulator configuration diagram.

The foldable 5-DOF manipulator designed in this paper is applied to EOD robots as shown in Figure 2 [13]. It contains 5 DOF, the manipulator arm span is up to 1.32 m and each joint is driven by a DMKE brushless servo motor. The clamping weight is 10 kg in the folded state, and the clamping weight is greater than or equal to 5 kg and less than 10 kg when fully expanded.

Considering the safety factors, the servo motor with brake function is selected as the driving device from the waist joint to the wrist rotation joint of the manipulator, which will stop at the original place after power failure. In addition, the ultrasonic sensor is used as the limit sensor to limit the angle of each link when the manipulator is reset, so as to realize the automatic reset function of the manipulator and ensure its operation stability. Considering many factors, we adopt a HC-SR04 ultrasonic sensor. Its ranging module can provide 2 cm–450 cm non-contact ranging function and ranging accuracy up to 3 mm. The module includes ultrasonic transmitter, receiver and control circuit.

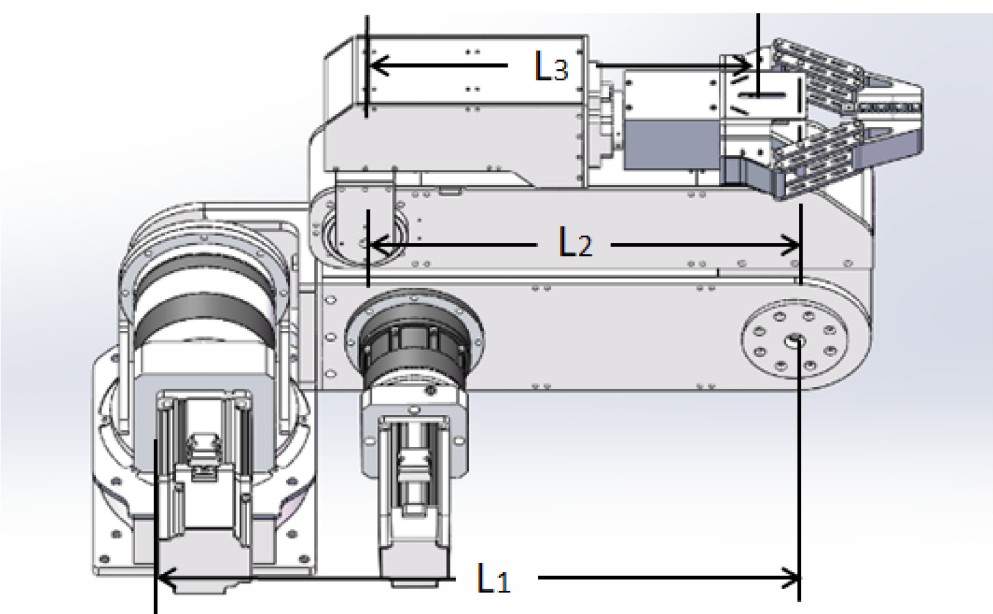

**Figure 2.** Prototype model diagram of the robot manipulator.

### 3. Kinematics Analysis of 5-DOF Manipulator

*3.1. Construction of D–H Model and Parameter Table*

The research object is a manipulator with 5 rotational DOF. Due to the complexity of the mechanical arm structure, there are many bends and offsets between each joint axis. The bending occurs at the joints of the manipulator. If the traditional D–H parameter table is used to build the forward kinematics model, it will be difficult to measure the parameters accurately. The problem is that when the machining or installation accuracy is insufficient, the adjacent parallel joints will be close to parallel with a certain angle. Therefore, the virtual joint is introduced; that is, an unchangeable joint is added between the actual two joints to change the need to measure when building the D–H parameter table, so as to reduce the difficulty of parameter measurement and improve the modeling accuracy. After introducing the virtual joint, the manipulator model has six joints, one of which is located between the elbow joint and the wrist pitch joint. The structure diagram and the establishment of coordinate system after the introduction of virtual joint are shown in Figure 3.

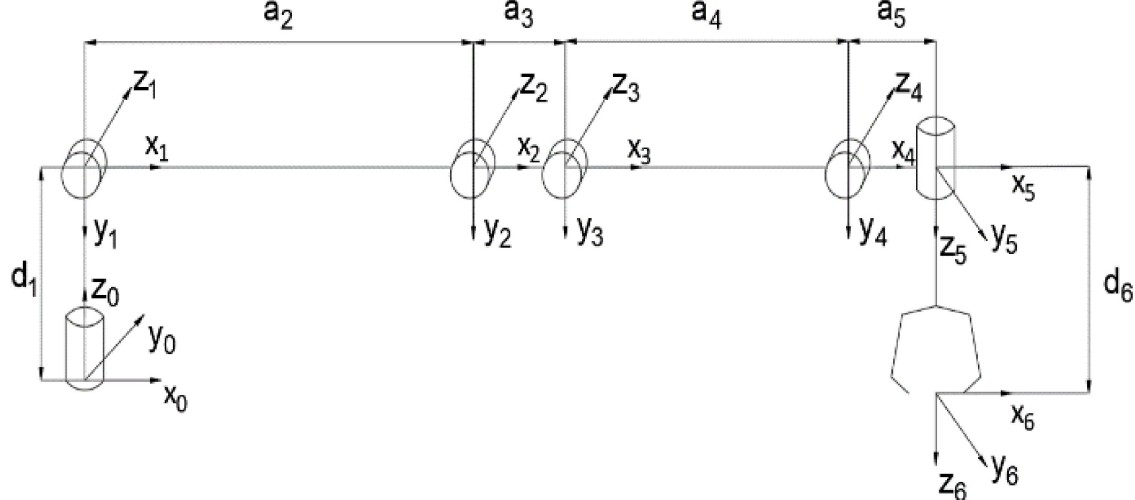

**Figure 3.** Structure sketch of the manipulator containing a virtual joint.

Based on the coordinate system of each joint in the figure above, the D–H parameter table can be obtained, as shown in Table 1. In the table, $d_i$, $a_i$, $\alpha_i$ and $\theta_i$ represent the joint offset, connecting rod length, connecting rod angle and joint angle, respectively [14].

**Table 1.** The D–H parameter table of manipulator after introducing virtual joint.

| Joint | $d_i$ | $a_i$ | $\alpha_i$ | $\theta_i$ | Joint Range/° |
|---|---|---|---|---|---|
| Waist joint | $d_1$ | 0 | Pi/2 | $\theta_1$ | 0~180 |
| Shoulder joint | 0 | $a_2$ | 0 | $\theta_2$ | −60~180 |
| Elbow joint | 0 | $a_3$ | 0 | $\theta_3$ | −120~90 |
| Virtual joint | 0 | $a_4$ | 0 | $\theta_4$ | 90 |
| Wrist pitch joint | 0 | $a_5$ | Pi/2 | $\theta_5$ | −90~90 |
| Wrist rotation joint | $d_6$ | 0 | 0 | $\theta_6$ | −180~180 |

### 3.2. Forward Kinematics Analysis

In the forward kinematics analysis, the joint angle variable $\theta_i$ is known for each joint of the robot, and the position and attitude of the robot end are calculated. According to the coordinate transformation rule of the manipulator, the relationship between the two connecting rods can be represented by the homogeneous transformation matrix:

$$
^{i-1}T_i = \begin{bmatrix} \cos\theta_i & -\sin\theta_i\cos\alpha_i & \sin\theta_i\cos\alpha_i & a_i\cos\theta_i \\ \sin\theta_i & \cos\theta_i\cos\alpha_i & -\cos\theta_i\sin\alpha_i & a_i\sin\theta_i \\ 0 & \sin\alpha_i & \cos\alpha_i & d_i \\ 0 & 0 & 0 & 1 \end{bmatrix} \tag{1}
$$

The homogeneous transformation of each joint can be derived from Equation (1) and Table 1. To be concise, in the following, $\sin\theta_i$ and $\cos\theta_i$ are denoted by $C_i$, and $\sin(\theta_m + \theta_n)$ and $\cos(\theta_m + \theta_n)$ are denoted by $S_{mn}$ and $C_{mn}$, respectively, where $i, m, n \in \{0, 1, 2, 3, 4, 5, 6\}$.

$$
^{0}T_1 = \begin{bmatrix} C_1 & 0 & S_1 & 0 \\ S_1 & 0 & -C_1 & 0 \\ 0 & 1 & 0 & d_1 \\ 0 & 0 & 0 & 1 \end{bmatrix} \tag{2}
$$

$$
^{1}T_2 = \begin{bmatrix} C_2 & -S_2 & 0 & a_2C_2 \\ S_2 & C_2 & 0 & a_2S_2 \\ 0 & 0 & 1 & 0 \\ 0 & 0 & 0 & 1 \end{bmatrix} \tag{3}
$$

$$
^{2}T_3 = \begin{bmatrix} C_3 & -S_3 & 0 & a_3C_3 \\ S_3 & C_3 & 0 & a_3S_3 \\ 0 & 0 & 1 & 0 \\ 0 & 0 & 0 & 1 \end{bmatrix} \tag{4}
$$

$$
^{3}T_4 = \begin{bmatrix} C_4 & -S_4 & 0 & a_4C_4 \\ S_4 & C_4 & 0 & a_4S_4 \\ 0 & 0 & 1 & 0 \\ 0 & 0 & 0 & 1 \end{bmatrix} \tag{5}
$$

$$
^{4}T_5 = \begin{bmatrix} C_5 & 0 & S_5 & a_5C_5 \\ S_5 & 0 & -C_5 & a_5S_5 \\ 0 & 1 & 0 & 0 \\ 0 & 0 & 0 & 1 \end{bmatrix} \tag{6}
$$

$$
^{5}T_6 = \begin{bmatrix} C_6 & 0 & S_6 & 0 \\ S_6 & 0 & -C_6 & 0 \\ 0 & 1 & 0 & d_6 \\ 0 & 0 & 0 & 1 \end{bmatrix} \tag{7}
$$

By multiplying the above Equation (2) to Equation (7), the position and attitude of the end-effector center of the manipulator can be obtained. See Equation (8).

$$
{}^{0}T_{6} = {}^{0}T_{1}{}^{1}T_{2}{}^{2}T_{3}{}^{3}T_{4}{}^{4}T_{5}{}^{5}T_{6} = \begin{bmatrix} n_x & o_x & a_x & p_x \\ n_y & o_y & a_y & p_y \\ n_z & o_z & a_z & p_z \\ 0 & 0 & 0 & 1 \end{bmatrix}
\tag{8}
$$

By inputting the initial joint angle $\theta_i = 0°, (i = 1, 2, 3, 5, 6), \theta_4 = 90°$ into (8), the following can be obtained:

$$
{}^{0}T_{6} = \begin{bmatrix} 0 & 0 & 1 & 1.075 \\ 0 & -1 & 0 & 0 \\ 1 & 0 & 0 & 0.6410 \\ 0 & 0 & 0 & 1 \end{bmatrix}
\tag{9}
$$

The results are completely consistent with those obtained from the fkine forward kinematics solution function in the MATLAB Robotic Toolbox. Fkine is a function name of the software MATLAB Robotic Toolbox. At the same time, it is consistent with the D–H coordinate system created in this paper. It can be seen that the position and attitude vector of the end-effector center of the manipulator and the forward kinematics equation are correct.

### 3.3. Inverse Kinematics Analysis

The inverse kinematics is known to be the position and attitude of the robot end, and all the joint variables of the corresponding position of the robot are solved, namely, the joint angle $\theta_i$. In the field of engineering, inverse kinematics has a higher application value and is the basis of robot motion planning and trajectory planning [15]. To obtain this closed-form solution, two sufficient conditions need to be followed [16–18]:

(1)   Three adjacent joint axes intersect at one point;
(2)   There are three adjacent joint axes parallel to each other.

The manipulator designed in this paper satisfies the second condition: shoulder joint, elbow joint and wrist pitch joint are parallel to each other. Therefore, the closed solution of robot inverse kinematics can be obtained. For the transformation matrix of the end-effector of the manipulator, the inverse transformation method is used to list multiple equations, the analytical solution $\theta_4$ of the waist joint can be solved, and the analytical solutions of each subsequent joint can be obtained in turn.

$$
{}^{0}T_{6} = {}^{0}T_{1}{}^{1}T_{2}{}^{2}T_{3}{}^{3}T_{4}{}^{4}T_{5}{}^{5}T_{6} = \begin{bmatrix} n_x & o_x & a_x & p_x \\ n_y & o_y & a_y & p_y \\ n_z & o_z & a_z & p_z \\ 0 & 0 & 0 & 1 \end{bmatrix}
\tag{10}
$$

By arranging Equation (10) and pre-multiplying both sides of Equation (10) by ${}^{0}T_{1}{}^{-1}, {}^{1}T_{2}{}^{-1}, {}^{2}T_{3}{}^{-1}, {}^{3}T_{4}{}^{-1}, {}^{4}T_{5}{}^{-1}, {}^{5}T_{6}{}^{-1}$ in turn, one can obtain different equations [19], so as to obtain the value of each joint angle, as follows:

$$
\theta_1 = atan2(p_y, p_x)
\tag{11}
$$

$$
\theta_6 = atan2(n_x S_1 - n_y C_1, o_x S_1 - o_y C_1)
\tag{12}
$$

$$
\theta_{2345} = atan2(a_x C_1 + a_y S_1, -a_z)
\tag{13}
$$

$$
\theta_3 = atan2(a_3, a_4) - atan2\left(A^2 + B^2 - a_2^2 - a_3^2 - a_4^2, \pm\sqrt{\left[4a_2^2\left(a_3^2 + a_4^2\right) - \left(A^2 + B^2 - a_2^2 - a_3^2 - a_4^2\right)\right]}\right)
\tag{14}
$$

In the formula:

$$
A = p_x C_1 + p_y S_1 - a_5 C_{2345} - d_6 S_{2345}
$$

$$
B = p_z - d_1 - a_5 S_{2345} + d_6 C_{2345}
$$

$$
\theta_4 = pi/2
\tag{15}
$$

$$\theta_5 = atan2\left(\pm\sqrt{\left(M - (P^2 + Q^2 + X^2 + Y^2 - a_2^2)^2\right)}, P^2 + Q^2 + X^2 + Y^2 - a_2^2\right) - atan2(2 \times (Q \times Y - P \times X), 2 \times (P \times Y + Q \times X)) \quad (16)$$

In the formula:

$$N = n_z C_6 - o_z S_6$$
$$P = -(-a_5 a_z + d_6(a_x C_1 + a_y S_1) - p_x C_1 - p_y S_1).$$
$$Q = -\left(a_5(a_x C_1 + a_y S_1) + d_6 a_z - p_z + d_1\right)$$
$$X = a_3 a_z + a_4 N, \; X = a_3 a_z + a_4 N$$
$$M = 4 \times \left((Q \times Y - P \times X)^2 + (P \times Y + Q \times X)^2\right)$$
$$\theta_2 = \theta_{2345} - \theta_3 - \theta_4 - \theta_5 \quad (17)$$

As can be seen from (14) and (16), there may be $2^2$ joint combinations of the manipulator in the same position and orientation. However, due to the existence of the ultrasonic limiter on the manipulator, the value range of each joint variable is shown in Table 1. During the solution process, given the end target position and orientation pose T, only one group of four original solutions satisfies the range of joint variables, so the multiple solution optimization problem is not involved in the inverse kinematics solution process of the manipulator.

## 4. Manipulator Workspace Verification

In order to determine whether the manipulator can grasp the target and put it into the explosion-proof tank, the reachable workspace of the manipulator must be analyzed. The reachable workspace is a collection of all the positions that can be reached by the terminal center point of the robot. Based on the Monte Carlo pseudo-random number method, the manipulator model is established in MATLAB, and the extracting parameter is substituted in the forward kinematics equation of the manipulator. When the sample number of random sampling points is set as $1 \times 10^7$, the reachable workspace of the end-effector of the manipulator can be solved as shown in Figure 4.

As shown in Figure 4, the executable operation region of the end-effector of the manipulator covers all the space in the maximum accessible area, which can accurately reflect the operation space of the robot.

Figure 5 shows that the coordinates of the explosion-proof tank mouth center in the manipulator coordinate system are [0.53 m, −0.23 m, 0.06 m]. Obviously, the center of the explosion-proof tank mouth is in the workspace of the manipulator.

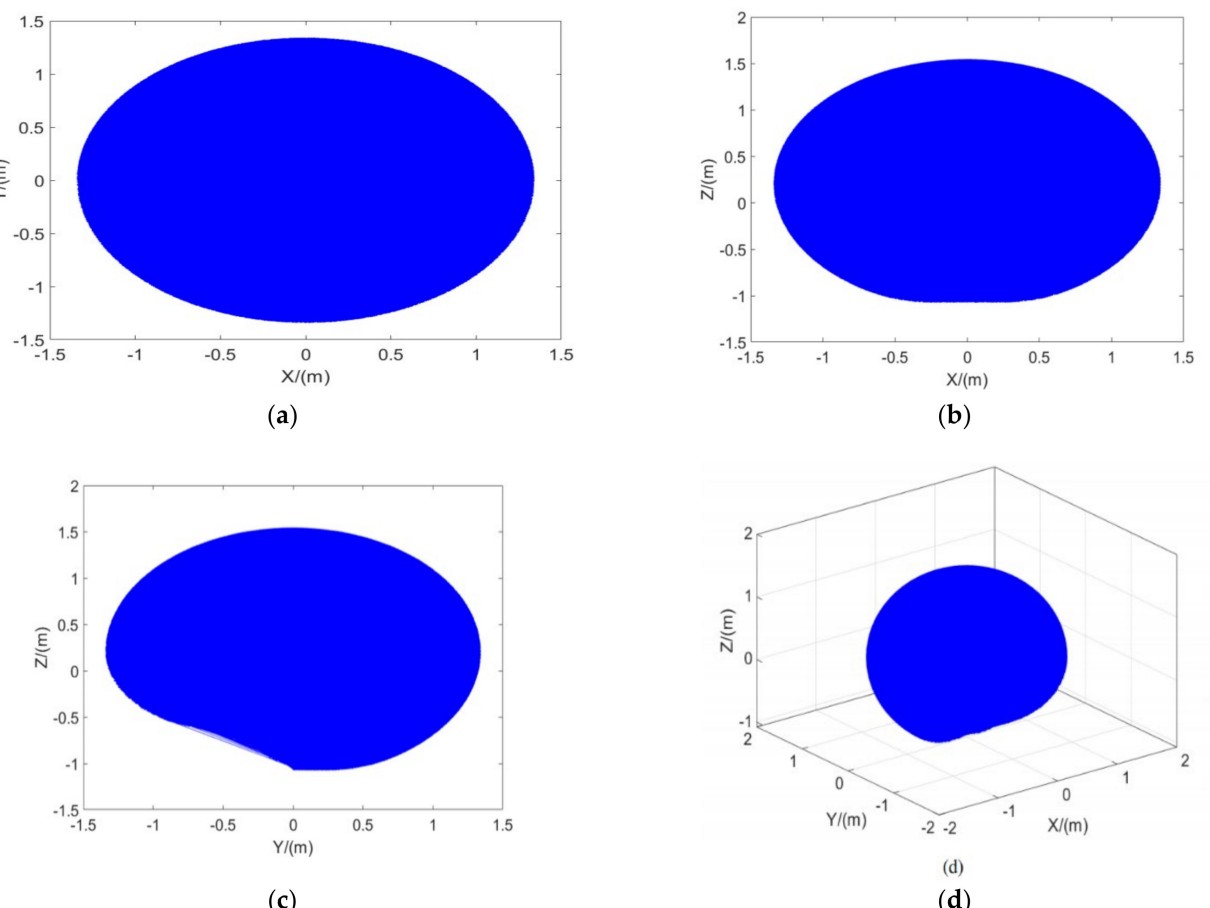

**Figure 4.** The reachable workspace simulation diagram: (**a**) The XOY coordinate plane projection drawing of workspace; (**b**) The XOZ coordinate plane projection drawing of workspace; (**c**) The YOZ coordinate plane projection drawing of workspace; (**d**) Isometric view of workspace.

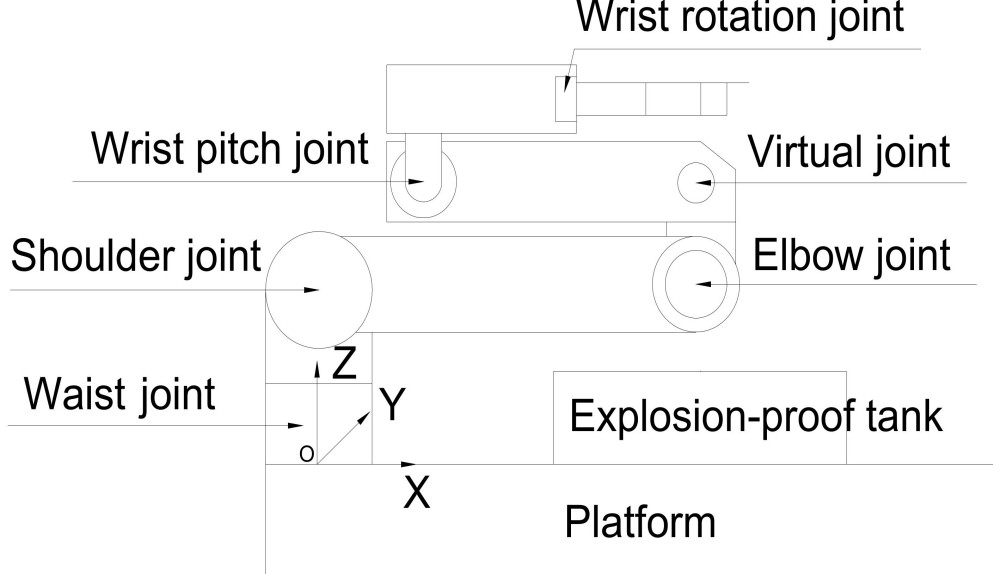

**Figure 5.** Schematic diagram of the coordinate system position of the manipulator in the center of the explosion-proof tank.

## 5. Simulation and Experiment

### 5.1. Simulation of Manipulator Trajectory Planning

The purpose of trajectory planning is to generate reference input of the motion control system by analyzing the desired motion path. The aim is to complete the path in the shortest time under certain constraints on factors such as the speed and acceleration of joints [20]. At present, trajectory planning is mainly divided into two aspects: one is multiple continuous points along the specified path, and the other is the starting and ending point of the specified path. In this paper, the working condition of the manipulator is to move between two points, so the second idea is chosen. Let the starting point of the manipulator be point A, as shown in Figure 6a, and the ending point of the manipulator be point B, as shown in Figure 6b. On the premise of determining the pose matrix of the two points A and B, the path point is converted into the angles $q_a$ and $q_b$ of each joint at two points through inverse kinematics analysis. Subsequently, the constraint conditions of each joint during the motion of the manipulator are considered comprehensively, so as to obtain the changing trajectory of the manipulator.

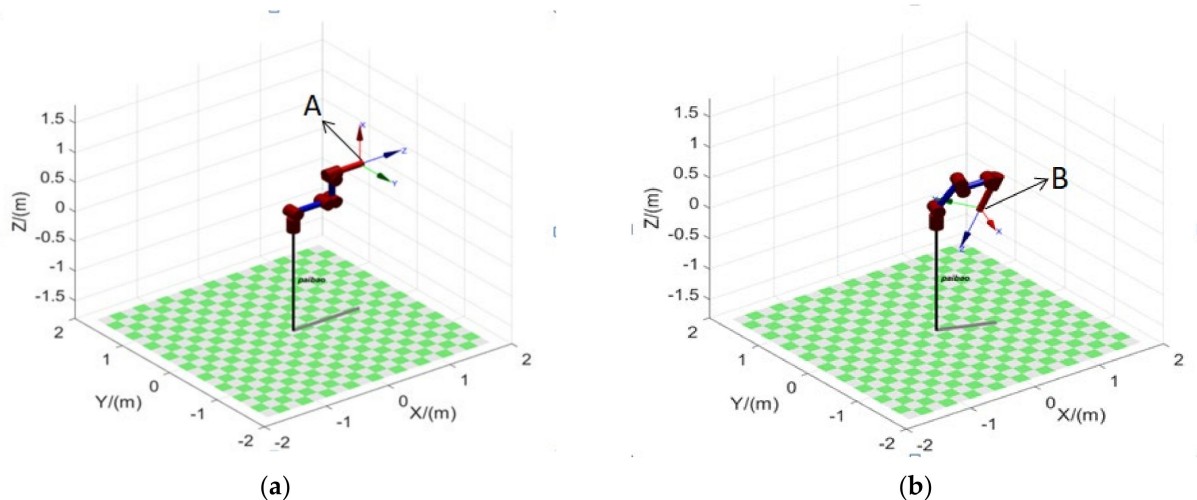

**(a)**                                                **(b)**

**Figure 6.** MATLAB simulation diagram: (**a**) Initial position and orientation; (**b**) Terminal position and orientation.

We use MATLAB Robotic Toolbox to simulate the trajectory of the manipulator. By calling the jtraj( ) command, the 5th order polynomial interpolation trajectory in joint space can be obtained. By bringing each joint angle $q_0 = [0, 0, pi/2, pi/2, -pi/2, 0]$ and $q_1 = [-0.117 * pi, 0.3 * pi, -0.728 * pi, pi/2, -0.202 * pi, pi/2]$ of the above starting and ending point into $[q \; qd \; qdd] = \text{itraj}(q_0, q_1, \text{step})$, the variation curves of each joint q, velocity qd and acceleration qdd of the manipulator are obtained, as shown in Figure 7a–c.

Figure 7a shows the change in angular displacement of each joint. Each joint angle smoothly changes from the initial angle to the terminal angle, and the virtual joint angle does not change. According to the inverse kinematics solution of the two points, it can be seen that the motion of each joint is correct. Figure 7b,c prove that the angular velocity and angular acceleration curves of each joint of the manipulator obtained by simulation are successive and smooth, which proves that each joint of the manipulator operates steadily during the movement, and there is no obvious catastrophe point. It meets the working requirements of the manipulator. The terminal trajectory of the manipulator is shown in Figure 8.

Figure 8 shows that the proposed algorithm can be used to move the EOD manipulator from the initial position to the target position through each expected point, which guarantees the functionality of the EOD manipulator and proves the reliability of the algorithm.

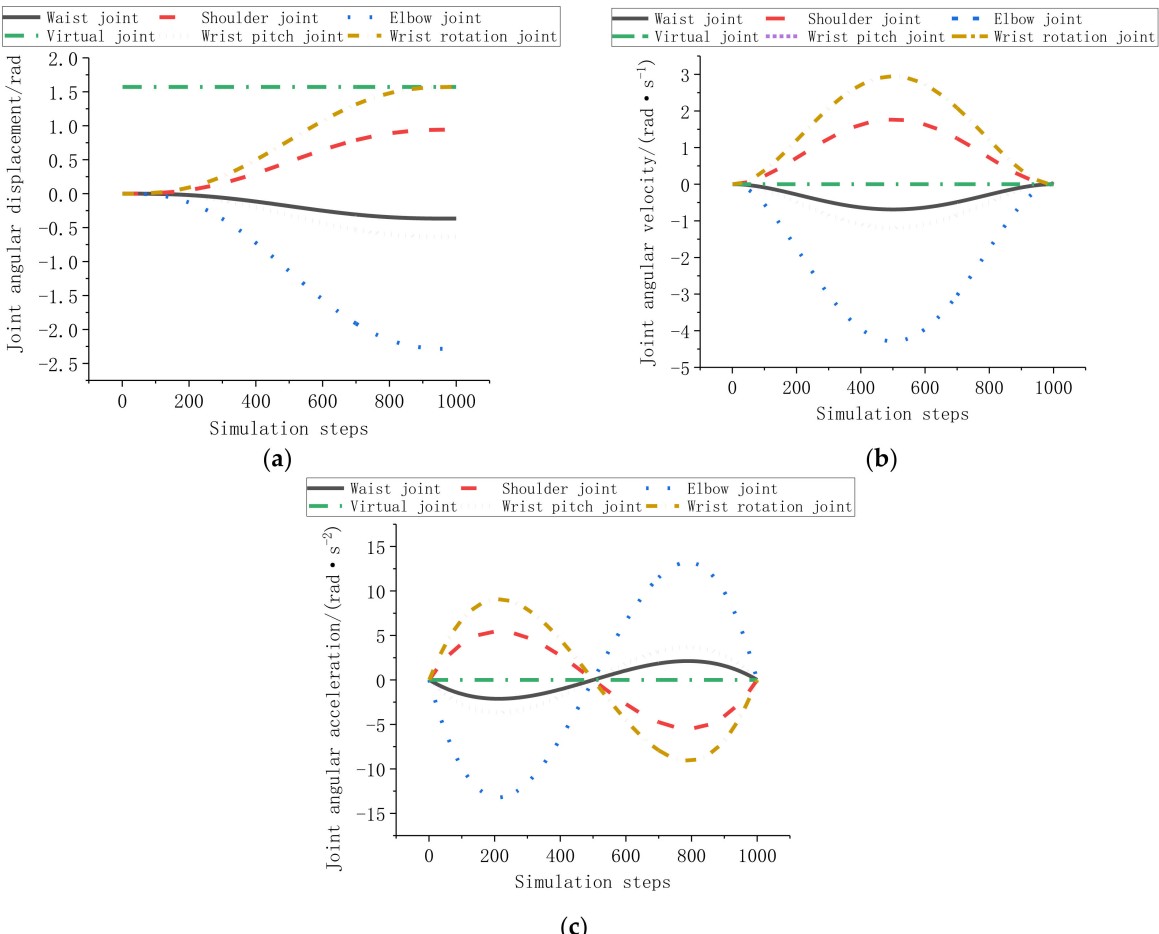

**Figure 7.** (**a**) Angular displacement curve of each joint; (**b**) Angular velocity curves of each joint; (**c**) Angular acceleration curves of each joint.

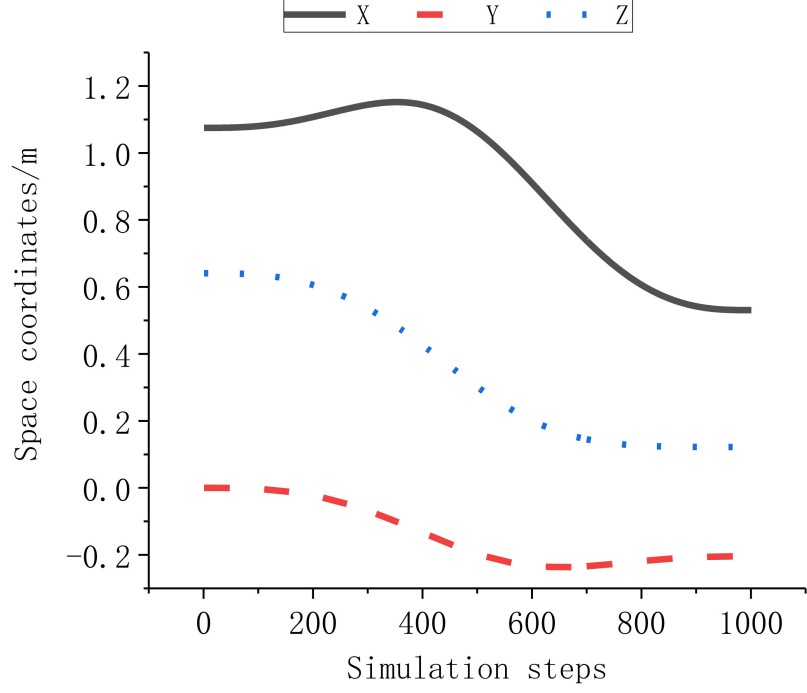

**Figure 8.** End displacement curve of the manipulator.

### 5.2. Robot Prototype Experiment

The manipulator is installed on the mobile robot, and the performance of the robot also directly affects the work of the manipulator. Therefore, we carried out experimental verification through the prototype to verify whether the actual performance of the explosive ordnance disposal (EOD) robot meets the requirements.

The main technical parameters of the mechanism are shown in Table 2.

**Table 2.** Main technical parameters of the mechanism.

| Technical Index | Parameter |
| --- | --- |
| Mobile platform size (mm) | $600 \times 1100 \times 900$ |
| Dead weight of mobile platform (kg) | $\leq 200$ |
| Mobile platform load (kg) | $\geq 100$ |
| Total mass (kg) | 20~35 |
| Platform load (kg) | $\leq 20$ |
| Motor speed(r/min) | 3000 |
| Motor output torque (N·m) | 7.78 |
| Battery life (h) | 4 |
| Climbing ability | $\geq 25°$ |
| Supply voltage (V) | 48 |

Figure 9 shows is the construction process of the EOD robot experimental prototype. The manipulator is installed on the EOD robot, and both the manipulator and EOD robot are controlled by the remote control box MCU, as shown in Figure 10.

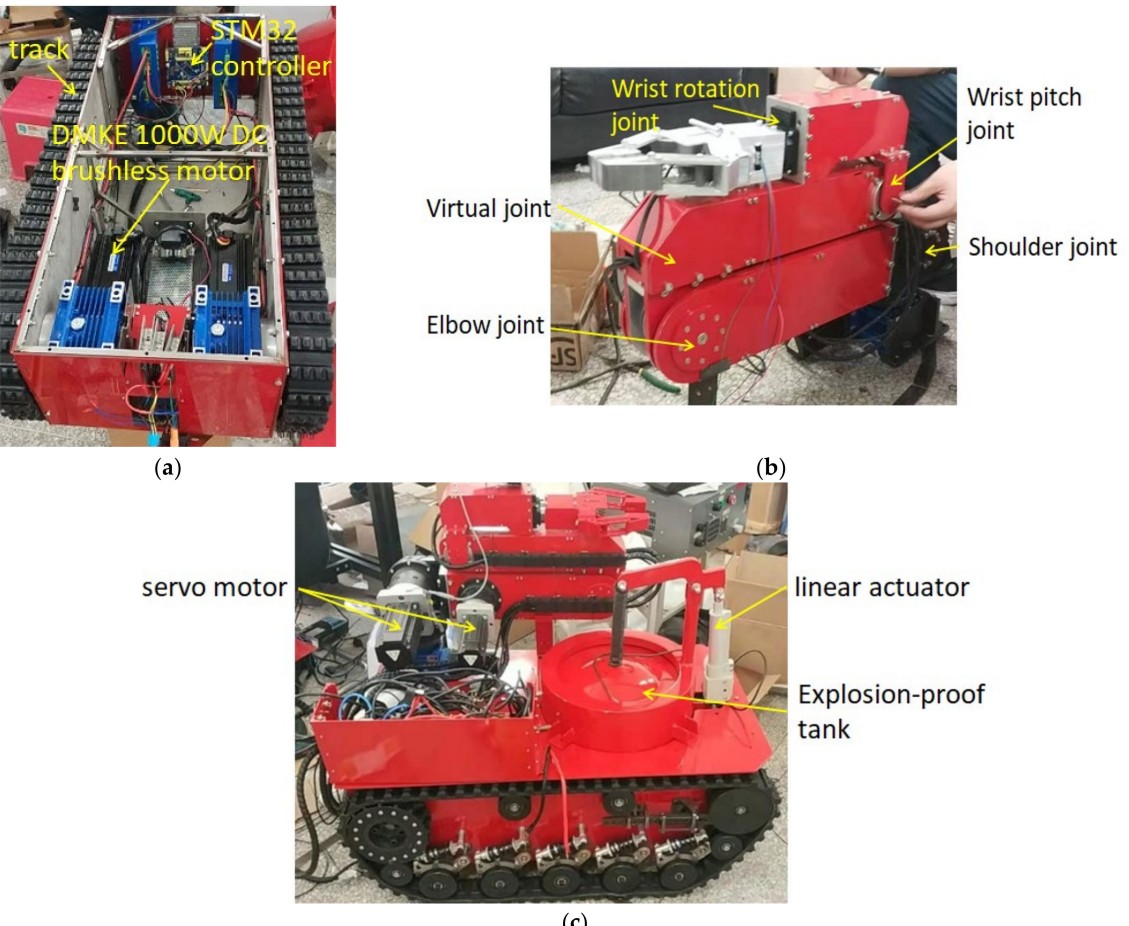

**Figure 9.** Construction process of experimental prototype: (**a**) Electrical components of mobile sites; (**b**) Components of mechanical arm; (**c**) Other components of EOD robot.

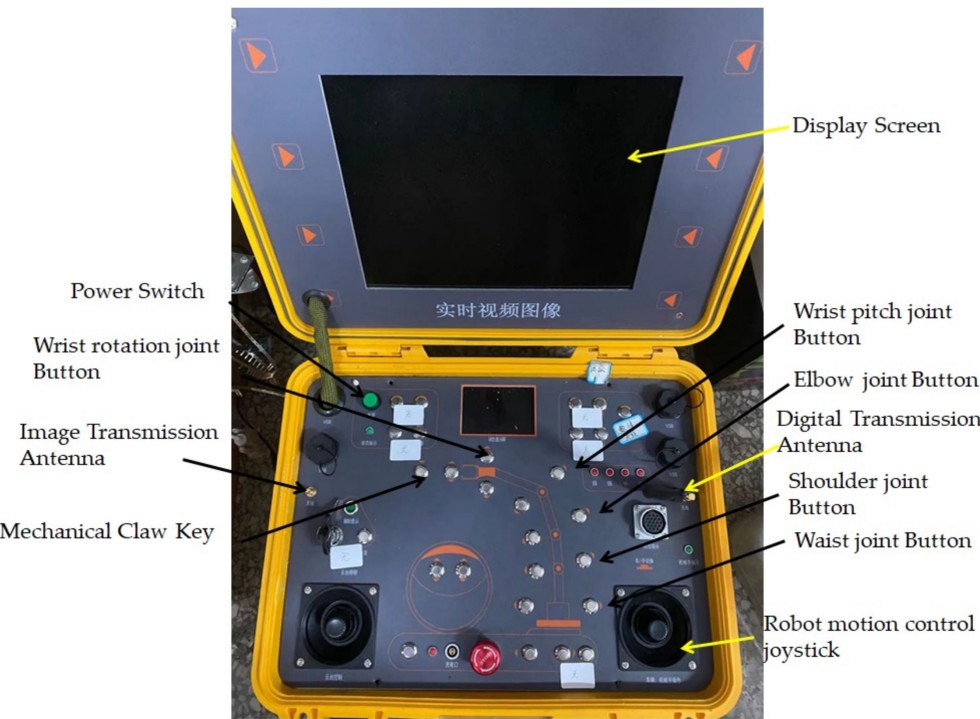

**Figure 10.** The remote control box MCU.

The EOD robot system consists of two parts, the remotely controlled box and the main body of the 5-DOF manipulator. The remotely controlled box can control the movement of the robot chassis. The system structure is shown in Figure 11.

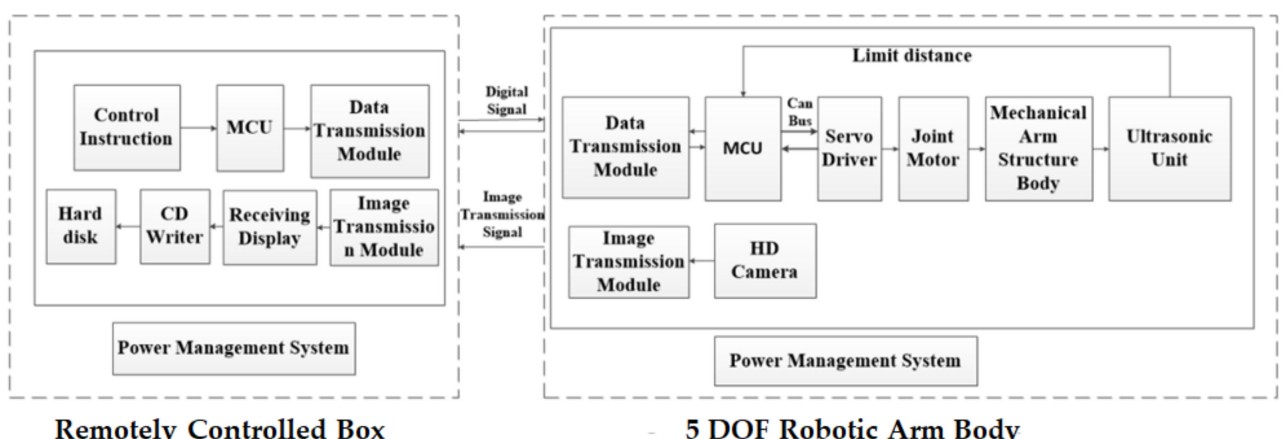

**Figure 11.** EOD Robot control system diagram.

Due to environmental constraints, simple steering and obstacle crossing experiments are only carried out under the existing conditions of the laboratory.

Figure 12 shows the robot in situ steering experiment. The steering performance of the explosive disposal robot has a significant impact on its traffic ability in narrow terrain. The speed of driving wheels on both sides are set to 0.3 m/s$^2$ for in situ steering performance test. During steering, although a large force is generated between the track and the ground, the tracks on both sides do not fall off. At this time, the radius of the driving wheel R = 0.1 m, which is almost in situ steering.

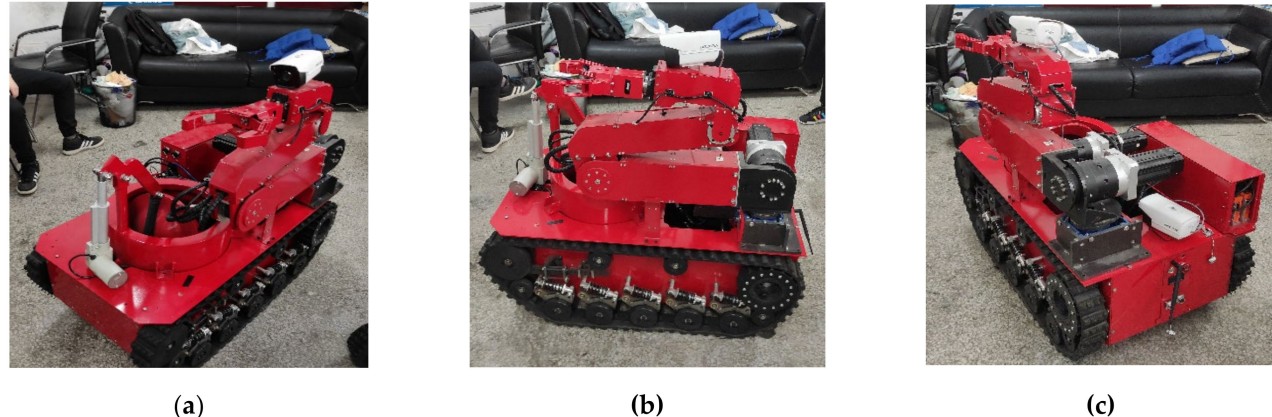

**Figure 12.** Robot in situ steering performance test: (**a**) Initial state of situ steering; (**b**) During situ steering; (**c**) In place intu steering end state.

Because the prototype experiment can complete almost in situ steering, the robot can complete turning in a narrow right angle space. The experimental results show that the robot can complete in situ steering and has good passing performance.

Multi-layer steps can be regarded as a combination of vertical obstacles and slopes. If the EOD robot can climb multi-layer steps with a certain angle, it also has the ability to climb slopes and climb vertical obstacles. As shown in Figure 13, the selected step has a height of 155 mm, a width of 365 mm and a slope of 25.2°, and the EOD robot can climb obstacles smoothly, which meets the design requirements and is within the required range. The specific analysis of obstacle crossing is shown in Figure 14.

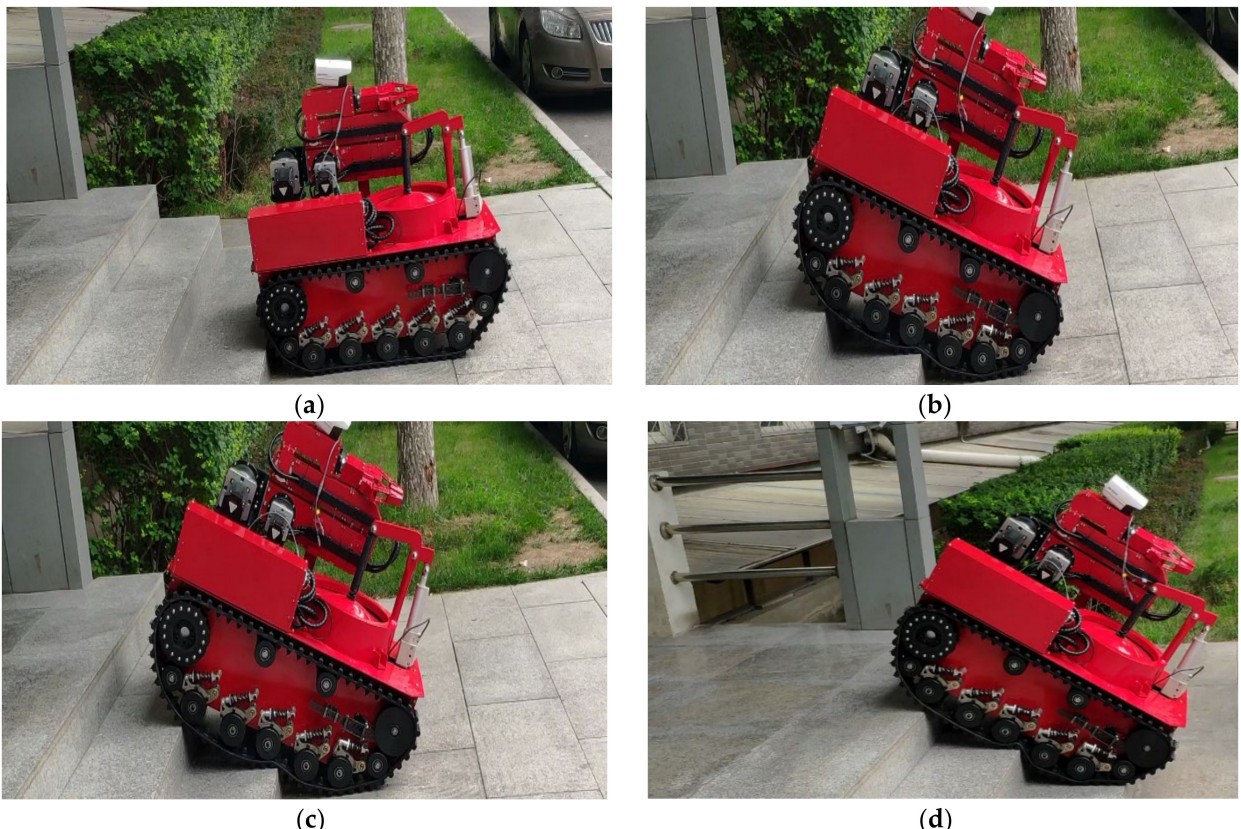

**Figure 13.** Obstacle crossing performance test of EOD robot: (**a**) Prepare for obstacle crossing; (**b**), (**c**) and (**d**) During EOD robot in obstacle crossing.

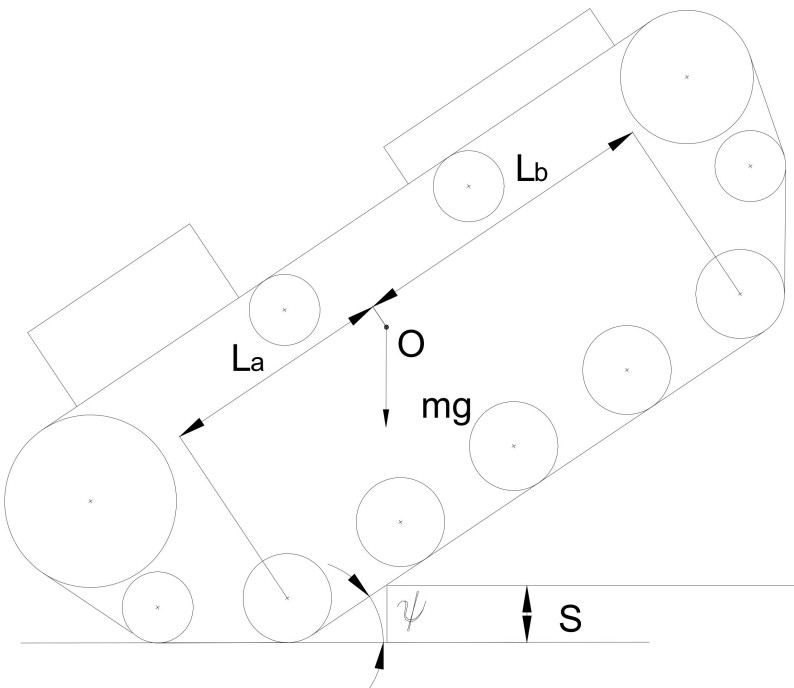

**Figure 14.** Robot vertical obstacle crossing limit position.

Figure 14 shows the limit state of robot vertical obstacle crossing. When the robot is in the critical position, the step height *S* equation can be obtained:

$$S(\psi, e, h) = \frac{L_c}{2} \sin \psi + e \sin \psi + R_4 (1 - \cos \psi) + h \left( \cos \psi - \frac{1}{\cos \psi} \right)$$

$\psi$ is the pitch angle of the robot;
$L_C$ is the length of the track connecting section;
$e$ is the longitudinal offset distance of the robot centroid.

MATLAB is used to obtain the relationship of $\psi, L_C$ and e when the pitch angle of the robot is 30°, as shown in Figure 15.

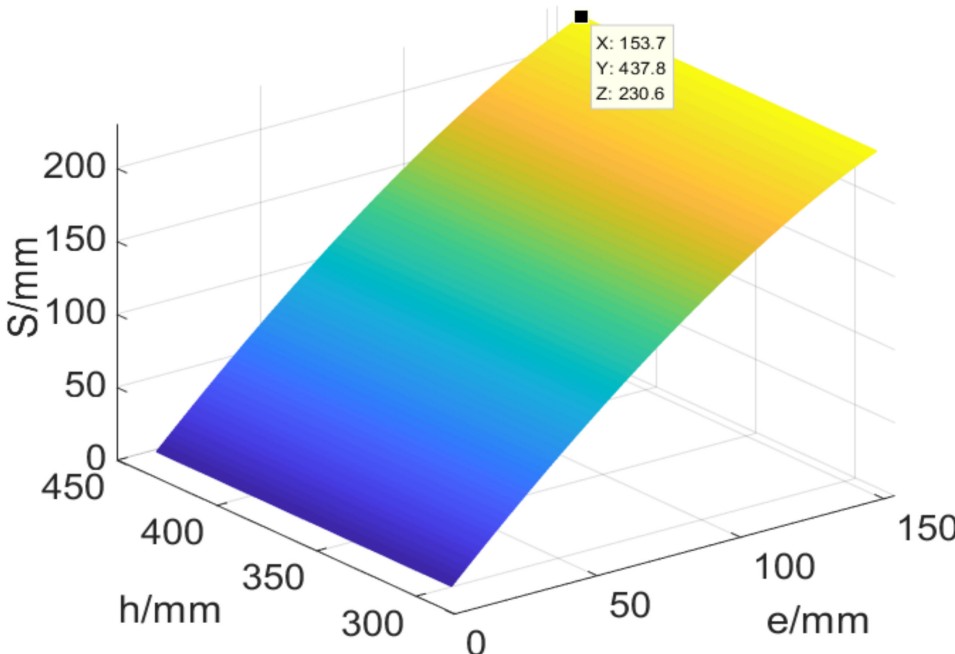

**Figure 15.** Relationship between obstacle crossing limit height and longitudinal position of centroid.

It can be seen from Figure 2 that the greater the longitudinal offset distance of the robot centroid, the higher the crossing obstacle height of the robot.

Therefore, it can be seen that the maximum height of the upper step $S_{\max}$ is:

$$S_{max} = min\{S(\psi, e, h), h''\}$$

### 5.3. Experimental Verification of Manipulator

The prototype is made to verify its running stability and grasping ability, and the trajectory planning process of grasping suspected explosives is carried out.

In order to ensure that the manipulator can execute the command accurately, the motion accuracy of each servo motor of the manipulator is adjusted before the experiment. Therefore, we introduce PID algorithm. The classical PID controller [21] can reduce the add-on error caused by the robot arm in the motion process and improve the accuracy of the robot arm joint tracking. This algorithm is used as the robot arm joint control strategy, as shown in Formula (18):

$$u(k) = K_p[e(k) + \frac{T}{T_i}\sum_{j=1}^{k} e(j) + \frac{T_d}{T}(e(k) - e(k-1))] \tag{18}$$

$$\Delta u(k) = u(k) - u(k-1) = K_p(e(k) - e(k-1)) + K_i e(k) + K_d(e(k) - 2e(k-1) + e(k-2)) \tag{19}$$

The meanings of the symbols in the above formula are shown in Table 3.

**Table 3.** Symbols and definitions.

| Symbol | Meaning |
| --- | --- |
| $u(k)$ | Computer output value at the k-th sampling time |
| $K_p$ | Proportional coefficient |
| $K_i$ | Integral coefficient |
| $K_d$ | Differential coefficient |
| $e(k)$ | Deviation value input at the k-th sampling time |
| $T$ | Sampling time |
| $T_i$ | Integration time |
| $T_d$ | Differential time |
| $k$ | Sampling serial number |

The diagram of wrist rotation joint motor control is shown in Figure 16 below. We obtain the desired value by adjusting various parameters of the motor, such as $K_p$, $K_i$ and $K_d$.

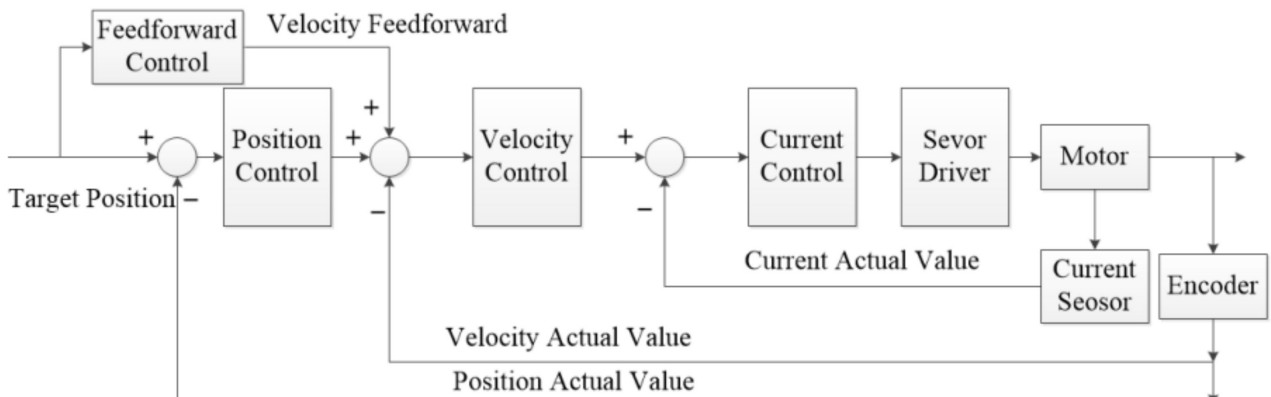

**Figure 16.** The diagram of wrist rotation joint motor control.

The change of the servo motor current loop before and after adjustment is shown in Figure 17.

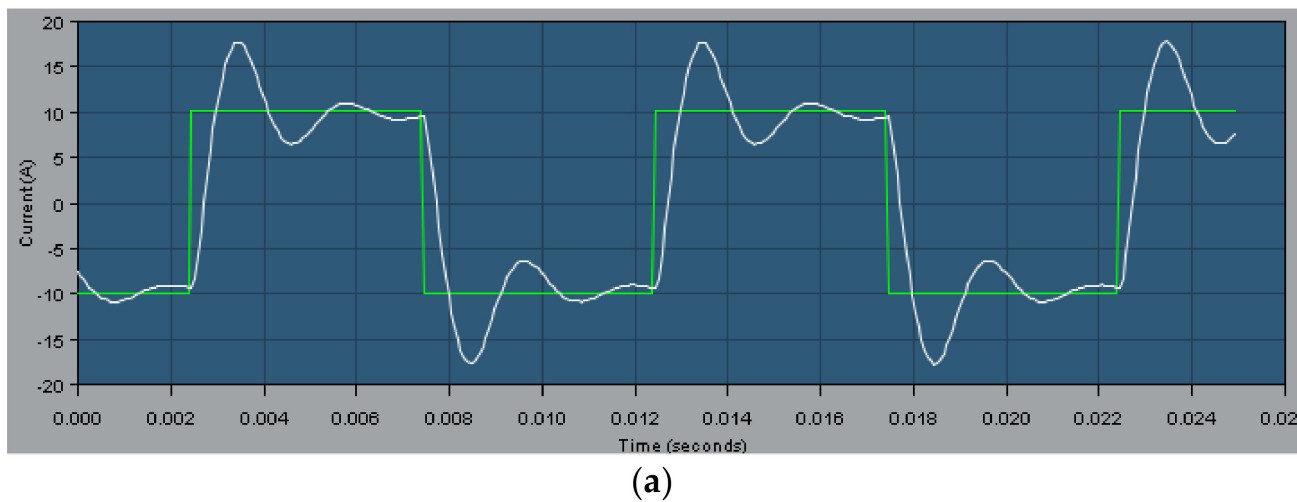

(a)

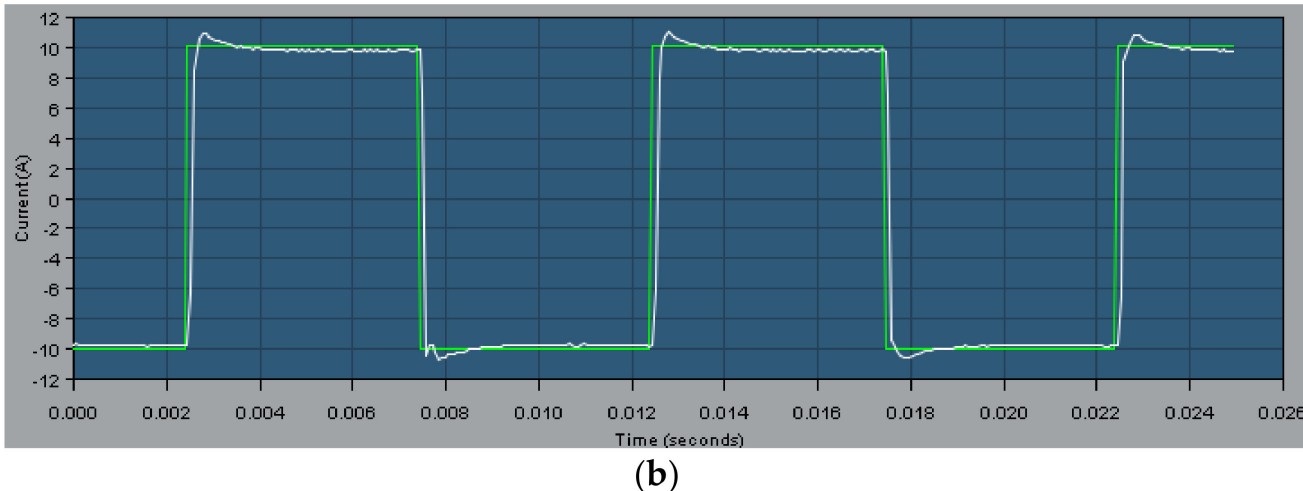

(b)

**Figure 17.** (**a**) Current change curve before debugging; (**b**) Current change curve after debugging.

As shown in Figure 17, the error between the actual current and the rated current is basically eliminated after adjustment, which meets the control requirements of the manipulator. The data from the starting point A to the ending point B obtained in the above simulation are applied to the actual motion control of the manipulator. Experimental process of manipulator trajectory planning is shown in Figure 18. By saving the data of the end-effector of the manipulator during the movement and using the data analysis software, the curve of the points passed can be drawn, as shown in Figure 19.

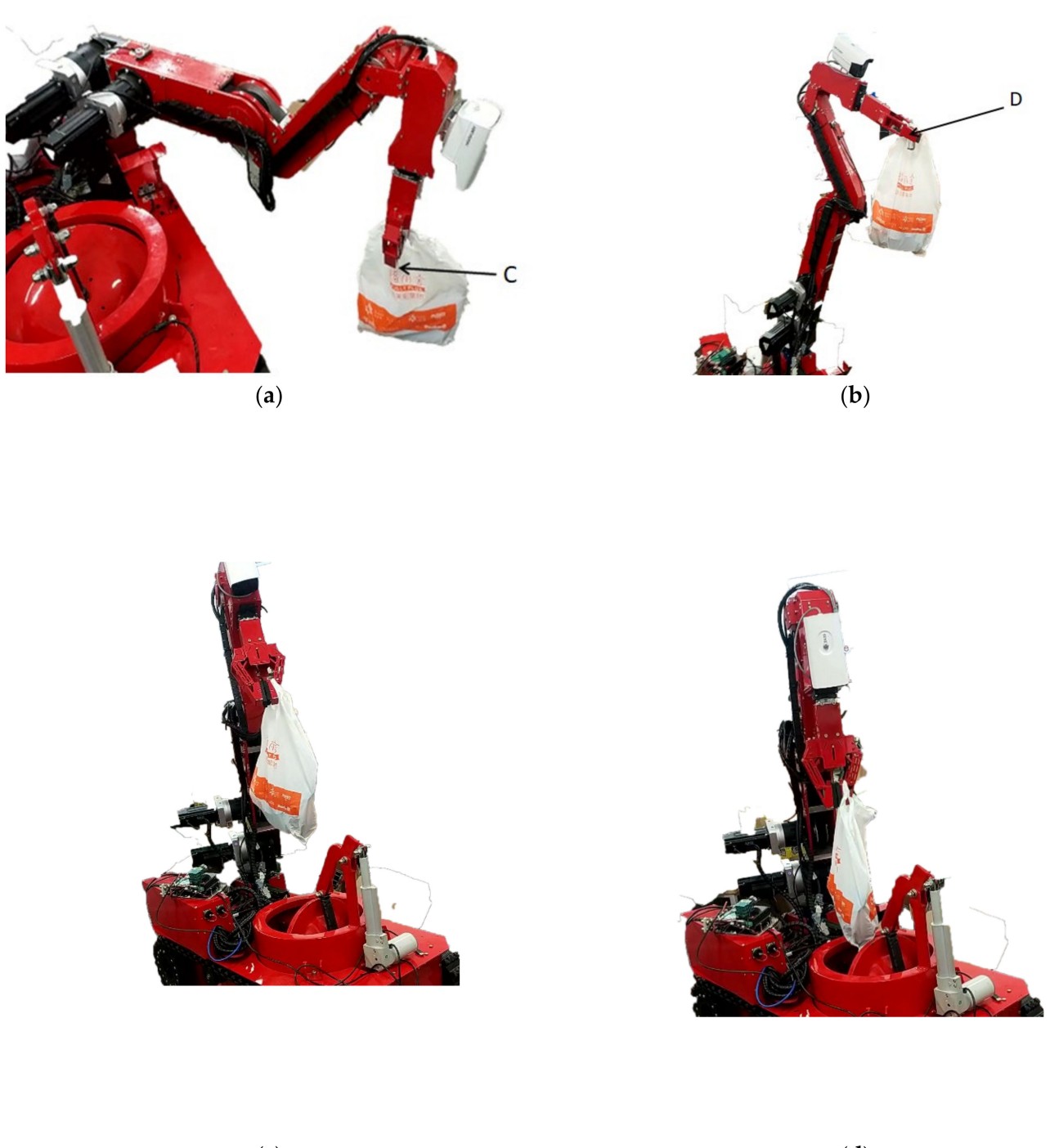

**Figure 18.** Experimental process of manipulator trajectory planning: (**a**) Control the mechanical claw to grasp the object; (**b**) The manipulator retracts the arm to the highest position; (**c**) Control the mechanical arm to the position above the explosion-proof tank; (**d**) Control the mechanical claw to grasp the object and put it into the explosion-proof tank.

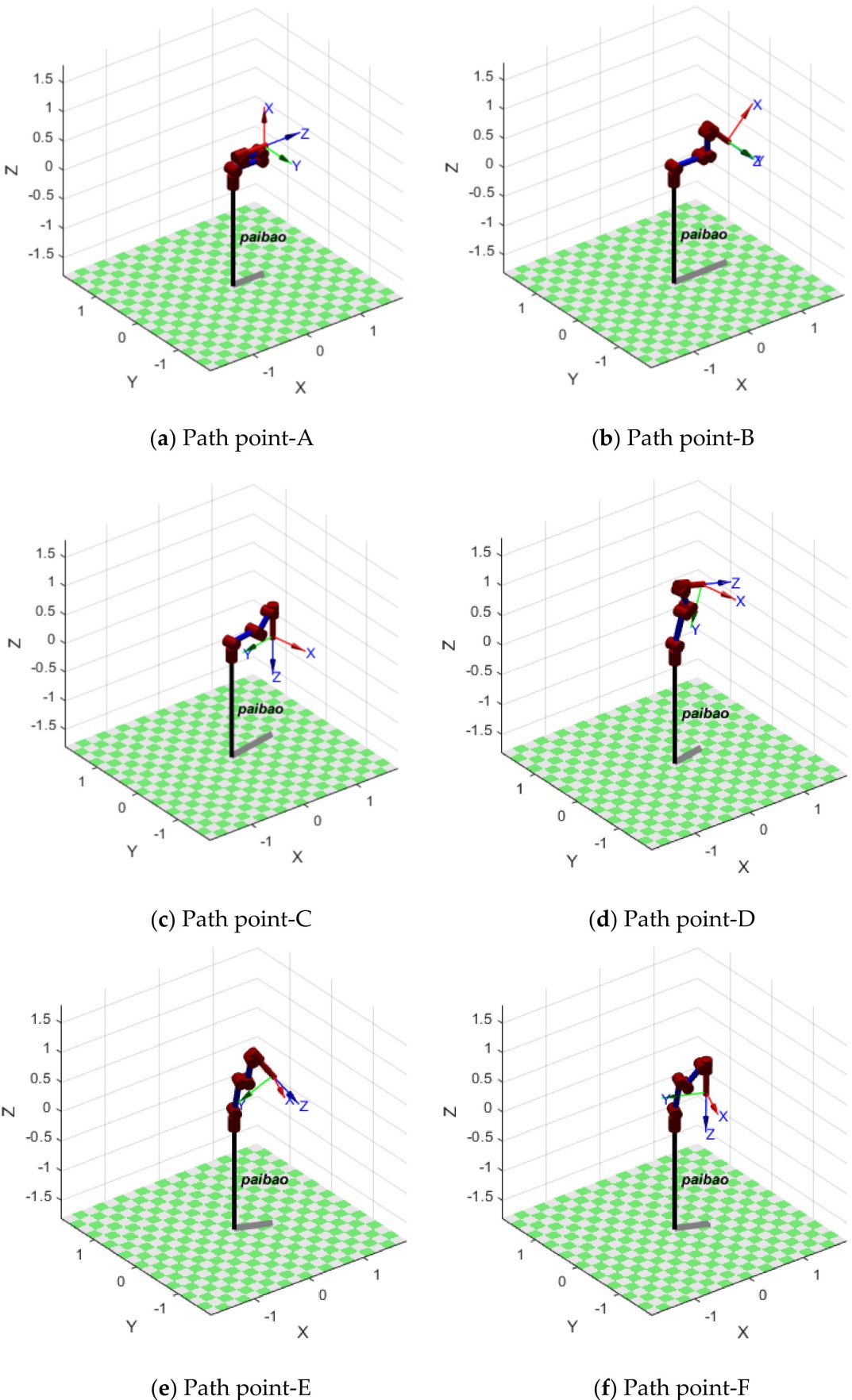

(**a**) Path point-A

(**b**) Path point-B

(**c**) Path point-C

(**d**) Path point-D

(**e**) Path point-E

(**f**) Path point-F

**Figure 19.** The simulated trajectory of each point of the manipulator.

Figure 18 shows that the motion process of the manipulator is stable and the running track is basically consistent with the simulation trajectory. Figure 20 shows the complete simulation path diagram. Comparing Figures 18 and 20, it can be seen that during the experiment, the manipulator can reach the planned middle path point. Figure a corresponds to path point C, and Figure 18b corresponds to path point D. The movement process is smooth without jitter, so it can complete the blasting task. Finally, it can be proved in combination with Figure 19 that the manipulator can reach the expected position and orientation when the starting and ending point of the path are specified and the path point is given, and the error can be basically ignored.

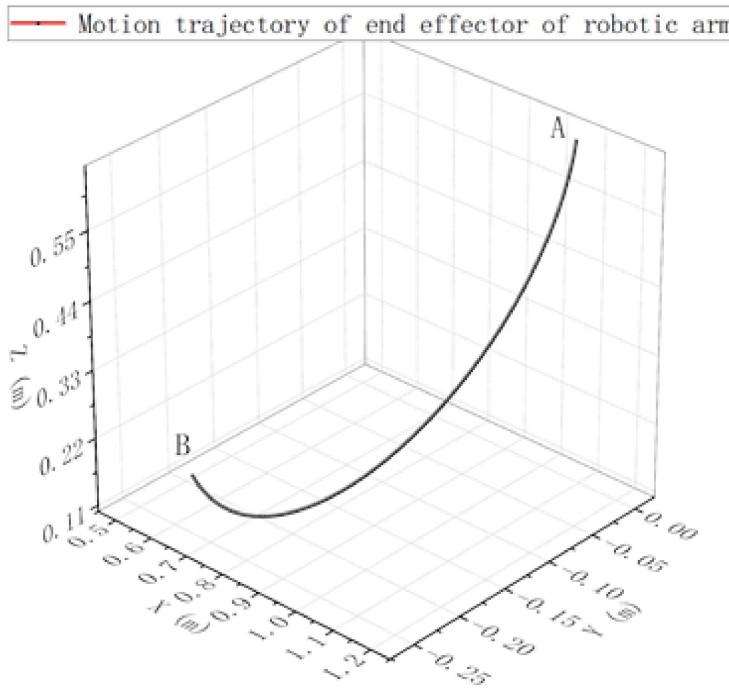

**Figure 20.** The actual trajectory of the manipulator.

Through the above simulation experiments and prototype experiments, the manipulator basically meets the design requirements. The experimental simulation and prototype experimental results show that the structural design of the manipulator is rational, ingenious, meets the design index of the manipulator and can realize the design requirements of placing explosives in the explosion-proof tank.

## 6. Conclusions

This paper proposes a foldable manipulator applied to the EOD robot, and the D–H parameter method is used to introduce the virtual joint, so as to establish the forward kinematics model of the manipulator. Based on the Monte Carlo pseudo-random number method, the workspace point cloud of the manipulator is given, which provides a reference for determining the actual activity space of the manipulator. The joint space trajectory planning of the manipulator is carried out by using 5th order polynomial interpolation functions, which provide the basis for the control of the manipulator. Finally, the actual prototype experiment is carried out by using the data obtained from the simulation, which verifies the effectiveness of the design and analysis of the manipulator.

In the future, an autonomous motion planning algorithm for the manipulator can be studied. After obtaining information on the suspected explosive through its vision system, it could grasp and generate the motion trajectory automatically, so as to realize autonomous intelligent explosive disposal.

**Author Contributions:** Conceptualization, J.Z. and T.H.; methodology, X.M.; software, T.H.; validation, T.H., C.L. and J.L.; formal analysis, T.H.; investigation, J.L.; resources, W.M.; data curation, T.H. and Y.L.; writing—original draft preparation, T.H.; writing—review and editing, J.Z. and T.H.; visualization, X.M.; supervision, J.Z.; project administration, J.Z.; funding acquisition, J.Z. All authors have read and agreed to the published version of the manuscript.

**Funding:** This research was supported in part by the National Social Science Foundation of China under Grant BIA200191.

**Institutional Review Board Statement:** Not applicable.

**Informed Consent Statement:** Not applicable.

**Data Availability Statement:** The data are available upon request.

**Acknowledgments:** The authors sincerely thank J.Z. of China University of Mining and Technology for his critical discussion and reading during manuscript preparation.

**Conflicts of Interest:** The authors declare no conflict of interest.

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
