# Peer review of "Research on Kinematics Analysis and Trajectory Planning of Novel EOD Manipulator"

_applsci, doi:10.3390/app11209438_

Round 1

Reviewer 1 Report

Now, some scientists have developed robots for dismantling explosive and burning devices. The British "trolley", series, the German "TEODOR" series and the American "Andros" series have been developed and applied in battlefields or terrorist activities. In China, there are also the "Lizards" series of EOD robots designed by the Institute of Automation of the Chinese Academy of Sciences, and the "Super-D" 46 series of EOD robots developed by Shanghai Jiao Tong University, equipped in 47 various subway stations, parks and other densely populated areas.

The article objectives are problems of mismatching poor flexibility and low precision of the ordinary manipulator in complex special explosive ordnance disposal process, so a new type of 5-DOF manipulator is proposed in this paper. The D-H model of the manipulator is established, and the forward and inverse kinematics are analyzed. The workspace of the manipulator is analyzed by Monte Carlo pseudo-random number method. MATLAB is used to simulate the trajectory planning, and the simulation data is used to verify the effectiveness of the design and analysis method.

The rest of this paper is arranged as follows: Section II briefly introduces the composition of the EOD robot and the design and analysis of the manipulator mechanism. Section III carries on the kinematics analysis to the manipulator. Section IV simulates its workspace. Section V introduces the trajectory planning, simulation and experimental verification of the manipulator.

This paper designs a foldable manipulator applied to the EOD robot, and the D-H parameter method is used to introduce the virtual joint, so as to establish the forward kinematics model of the manipulator. Based on the Monte Carlo pseudo-random number method, the workspace point cloud of the manipulator is given, which provides a reference for deter-mining the actual activity space of the manipulator. The joint space trajectory planning of the manipulator is carried out by using the 5th order polynomial interpolation functions, which provide the basis for the control of the manipulator. Finally, the actual prototype experiment is carried out by using the data obtained from the simulation, which verifies the effectiveness of the design and analysis of the manipulator.

In the future, autonomous motion planning algorithm for manipulator can be studied. After obtaining the information of the suspected explosive through the vision system, it can grasp and generate the motion trajectory automatically, so as to realize the autonomous intelligent explosive disposal.

I consider submitted article to be partly original.

More literary resources could be used in the present article.

The article would be appropriate in the end of the work explicitly indicate the scientific benefits of the proposed solutions.

At the end of the contribution, it is necessary to indicate the benefits of the proposed manipulator against so far. 

Author Response

Thanks to the editor for arranging you to review the manuscript for me, and thank you for your valuable comments. I will continue to study further in the follow-up research

Reviewer 2 Report

The manuscript proposes and investigates a 5-DOF folding manipulator. Adding DOF increases the flexibility of the manipulator but also increases the complexity and is discussed clearly in the manuscript.

The methods are appropriate for the problem the authors investigate. The results show a stable motion to the 5DOF and align well with existing literature. I find no faults in the paper and I recommend the paper for publication.

 - As a minor suggestion, the introduction of the study can be improved by adding more literature on the possibilities of the use of this 5-DOF manipulator as other than EOD robots.

- Line 195-196: "Let the starting point on manipulator be point A......." - I don't see Point A, B in Figure 6.

Author Response

Thanks to the editor for arranging you to review the manuscript for me, and thank you for your valuable comments.

Point : Line 195-196: "Let the starting point on manipulator be point A......." - I don't see Point A, B in Figure 6.

Response: Thank you for your reminder. I have made corrections in the Figure 6.

Reviewer 3 Report

I didn't find any significant novelty in your manuscript. 
There is not enough information about your design and your tests in the manuscript. There is not too much technical and scientific information nor engineering calculations. 
I have added more comments in attached the pdf file. 

Author Response

Thanks to the editor for arranging you to review the manuscript for me, and thank you for your valuable comments. I have carefully answered the questions one by one in accordance with your suggestions and requirements, and carefully revised the paper.

Round 2

Reviewer 3 Report

Thanks for your efforts in responding to my comments. I reviewed your manuscript carefully once again. I see that you have done a valuable job but this manuscript is not arranged appropriately and has major issues. The design part doesn't have enough explanation, especially the control part is missing. The manufacturing process of the robotic arm is not mentioned. For instance, the used material for the robotic arm, CAD figures, and so on are missed. If you add them, It can give your readers a better view and understanding of your robot. 

Most of the time you are not specific about what you are saying. You should talk to your readers clearly. For instance, you are saying the performance is good? How much good is it? Do you have any numbers and references proving that it is a good performance and meets the requirements? Probably something that is a good performance for you is very low for another robot or vice versa. Unfortunately, I cannot accept your manuscript but I think you can make it better.

I have added more comments in the pdf file. 

Good luck

Author Response

Thank you for your valuable comments. I have carefully answered the questions one by one in accordance with your suggestions and requirements, and carefully revised the paper.
Most of my responses are written under the peer review(applsci-1362051-review(revise)) questions you provided to me. I replied to your question accordingly.

Round 3

Reviewer 3 Report

Dear authors,

Thanks for paying attention to my comments. I tried to help you to put the manuscript in better shape. Now it looks better I think. 

The PID algorithm control block diagram that you provided is a very general diagram and looks redundant. Since this robot is doing a very crucial job its control should be very robust and reliable. A flowchart of robot processes may help to show the control process of the robot. It can help other researchers to have a better sense of your robot and take efficient steps to make their potential robots. 

Author Response

Thank you for your valuable comments. I have carefully answered the questions  in accordance with your suggestions and requirements, and carefully revised the paper.

I changed the  PID algorithm control block diagram  into a  wrist motor control block diagram to more clearly express  the control process of the robot.